# A secret from a hidden world: A new glassfrog of the genus *Nymphargus* (Anura: Centrolenidae) from Cordillera del Cóndor, Ecuador

**Mylena V. Masache-Sarango**[1�], **Diego F. Cisneros-Heredia**[2,3�], **Santiago R. Ron**[1�]*

**1** Museo de Zoología, Centro de Investigaciones de la Biodiversidad CIBIO, Facultad de Ciencias Exactas y Naturales, Pontificia Universidad Católica del Ecuador, Quito, Ecuador, **2** Museo de Zoología, Laboratorio de Zoología Terrestre, Instituto de Biodiversidad Tropical IBIOTROP, Colegio de Ciencias Biológicas y Ambientales, Universidad San Francisco de Quito USFQ, Quito, Ecuador, **3** Instituto Nacional de Biodiversidad INABIO, Quito, Ecuador

he These authors contributed equally to this work.
* santiago.r.ron@gmail.com

## Abstract

The genus *Nymphargus* is the most speciose of the family Centrolenidae with 44 species. In this study, we describe a new species of *Nymphargus* and present an updated phylogeny. The new species is sister to an undescribed species, also from SW Ecuador, and both belong to a clade that includes *N. buenaventura*, *N. cariticommatus*, *N. griffithsi*, *N. lasgralarias*, *N. sucre*, and *N. wileyi*. The new species likely originated during the Pliocene (~4.5 Mya) and is characterized by a uniformly green dorsum lacking spots, shagreened dorsal skin, and white peritonea covering the esophagus and stomach. Our phylogeny provides, for the first time, the phylogenetic position of *N. buenaventura*. The new species was discovered at Reserva Biológica El Quimi, during expeditions by the QCAZ Museum in 2017 and 2018. Most amphibian species found at that location were undescribed, indicating that some regions of Cordillera del Cóndor host amphibian communities that have remained as "hidden worlds" for biological exploration.

## Introduction

Glass frogs (family Centrolenidae) include 12 genera and about 167 species distributed throughout tropical America [1,2]. Most centrolenids have been included in morphological and molecular studies, and the phylogenetic relationships within the family are relatively well resolved [3]. However, several species remain undescribed or remain confused with other taxa, with many of these lineages represented by misidentified or unstudied museum collections [3,4].

The most speciose centrolenid genus is *Nymphargus* [5], with 44 species distributed across the tropical Andes, from Colombia to Bolivia, generally above 1000 m elevation [1–3,5–8]. There are 21 species of *Nymphargus* known from Ecuador,

**Data availability statement:** All newly generated sequences are available in GenBank under the accession numbers listed in Table 1. They are also available as part of the aligned matrix used for the phylogenetic analysis in Zenodo: https://zenodo.org/records/13323248.

**Funding:** Field and laboratory work in Ecuador was funded by Secretaría Nacional de Educación Superior, Ciencia, Tecnología e Innovación del Ecuador SENESCYT (Arca de Noé initiative; SRR). URL funding agency: https://www.educacionsuperior.gob.ec/ The funders had no role in study design, data collection and analysis, decision to publish, or preparation of the manuscript.

**Competing interests:** The authors have declared that no competing interests exist.

11 of which are endemic to the country [3,5,9–19]. Taxonomy in *Nymphargus* is challenging due to the high phenotypic similarity among many species. Additionally, some geographic regions remain poorly sampled, adding uncertainty to the distributional ranges and morphological variation of their species [3,5,6,8,9,12,14].

Cordillera del Cóndor is a Subandean mountain range separated from the Eastern Cordillera of the Andes of Ecuador by the Nangaritza–Bomboiza–Zamora River valleys. It extends along ca. 150 km north-south across the border between southeastern Ecuador and northern Peru, reaching a maximum elevation of ca. 3000 m [20]. Due to its remoteness and complex geology and geography, Cordillera del Cóndor holds unique biodiversity, including at least 23 endemic species of frogs: *Centrolene condor* [21]; *Excidobates condor* [22]; *Hyloxalus mystax* [23]; *Hyloscirtus condor* [24]; *H. hillisi* [25]; *Nymphargus colomai* [26]; *N. lindae* [3]; *Hyloscirtus maycu* [27]; *Osteocephalus duellmani* [28]; *Lynchius simmonsi* [29]; *Phyllonastes arutam* [30], *Phyllonastes plateadensis* [31]; *Pristimantis barrigai* [32]; *P. daquilemai* [33]; *P. ledzeppelin* [34]; *P. minimus* [35]; *P. muranunka* [36]; *Pristimantis muscosus* [37]; *P. nangaritza* [38]; *P. nanus* [39]; *P. paquishae* [40]; *P. pramukae* [39]; and *P. yantzaza* [41]. During expeditions to the Reserva Biológica El Quimi in the Cordillera del Condor, teams from the Museum of Zoology of Pontificia Universidad Católica del Ecuador collected specimens of *Nymphargus* initially misidentified as *N. cariticommatus* [19]. We integrate morphological and genetic analyses of this population to determine its taxonomic status. Our findings reveal that it represents a new species, which we describe below. For the first time, we also report the phylogenetic relationships of *Nymphargus buenaventura*, a species endemic to southwestern Ecuador.

## Materials and methods

### Ethics statement

Fieldwork and genetic sampling were authorized under a framework contract for access to genetic resources (MAAE-DBI-CM-2022–0230) and collecting authorizations (MAE-DNB-CM-2015–0025 and 011–2018-IC-FAU-DNB/MA) issued by the Ministerio del Ambiente, Agua y Transición Ecológica del Ecuador to Pontificia Universidad Católica del Ecuador. We followed the standard guidelines for using live amphibians in field research [42]. Protocols of collection and animal care comply with the Specific Regulation of the Animal Research Ethics Committee of Pontificia Universidad Católica del Ecuador (RE-CEIA-01).

### Species concept

We adopt the evolutionary species concept, defining a species as "a single lineage of ancestor-descendant populations of organisms which maintains its identity from other such lineages, and which has its own evolutionary tendencies and historical fate" [43]. We assumed that a lineage shows independent tendencies and historical "fate" (i.e., its own evolutionary trajectory) if genetic and morphological characters (independent data sets) covary and show a level of differentiation characteristic of interspecific differences in the genus *Nymphargus*. In morphology, these differences

typically assess the status of 23 diagnostic characters, the most important of which are external morphology, dorsal skin coloration, and internal anatomy [5].

## Fieldwork

Surveys were conducted on the sandstone table-top plateau on the eastern side of the Quimi River Valley, Reserva Biológica El Quimi, Morona Santiago province, Ecuador, in 2017. Live individuals were photographed and euthanized using topical 20% benzocaine. Muscle tissue samples were extracted and preserved in 95% ethanol; voucher specimens were fixed in 10% formalin and preserved in 75% ethanol. Specimens are deposited in the Museo de Zoología, Pontificia Universidad Católica del Ecuador, Quito (QCAZ).

## Taxonomic sampling

We examined specimens from The Natural History Museum, London (BMNH); División de Herpetología, Instituto Nacional de Biodiversidad, Quito (DHMECN); Instituto de Ciencias Naturales, Universidad Nacional de Colombia, Bogotá (ICN); University of Kansas Natural History Museum, Lawrence (KU); Museum of Comparative Zoology, Harvard University, Cambridge (MCZ); Museo de Zoología, Pontificia Universidad Católica del Ecuador, Quito (QCAZ); National Museum of Natural History, Smithsonian Institution, Washington, D.C. (USNM), Museo de Zoología, Universidad San Francisco de Quito, Quito (ZSFQ). Information on species for comparative diagnoses was obtained from the direct study of specimens and the literature [3,15,16,18,19,26,44–48]. Examined specimens are listed in S1 Appendix. Additional specimens examined during our studies in Centrolenidae are listed in [5] and [3].

## Morphology and coloration

Diagnosis, terminology and measurements follow the format and definitions proposed by Cisneros-Heredia & McDiarmid [5]. All morphological and morphometric data were obtained from adult specimens. Sex and reproductive status were determined by direct examination of the gonads through dissections and by noting the presence of secondary sexual characters. The following measurements were taken with a digital caliper (0.05 mm accuracy, rounded to nearest 0.1 mm), under a stereomicroscope, reported as a range (mean ± standard deviation), and are as follows: Snout–vent length (SVL), head width (HW), head length (HL), horizontal eye diameter (ED), inter–orbital distance (IOD), eye-nostril distance (EN), internarial distance between nostrils (IND), Finger III disc width (Fin3DW), tibia length (TL), foot length (FL), tympanum diameter (TD), length of finger I and finger II (FLI and FLII). The measurement methodology was based on [5]. Descriptions of the coloration in life are based on photographs taken in the field. The adjective "enamelled" describes the shiny white coloration produced by iridophore accumulation [5,16].

## Bioacoustic analyses

Calls were recorded with a Sennheiser™ ME-66 directional microphone and Olympus LS-14 recorder. Recordings were analyzed in Raven 1.6 [49] with a Hanning window function, 3 dB filter bandwidth of 135 Hz, DFT size 4096, frequency resolution 11.7 Hz; the time grid had 50% of overlap and a hop size of 5.33 ms. The sampling frequency was 48.0 kHz. Temporal variables were measured from oscillograms; spectral properties were measured from power spectra. Dominant frequency was measured along the duration of the complete call. Original recordings are deposited in the audio archive of the QCAZ collection and are available through the portal Bioweb.bio. The analyzed call is the only one associated with specimen QCAZ-A 68524. Ambient temperature is unavailable. Acoustic terminology follows the call-centered framework of Köhler et al. [50].

## Phylogenetic analysis

We generated DNA sequences in the molecular laboratory at QCAZ-A and combined them with sequences deposited in GenBank to obtain a new phylogeny of *Nymphargus*. We obtained DNA sequences of the mitochondrial genes 12S ribosomal RNA (12S), 16S ribosomal RNA (16S), and NADH dehydrogenase subunit 1 (ND1). We also sequenced the nuclear genes for the cellular myelocytomatosis gene (C-MYC) and recombination-activating genes (RAG1). Newly generated sequences comprise three individuals of the new species (QCAZ-A 68522, QCAZ-A 68526 and QCAZ-A 68586), *Nymphargus buenaventura* (QCAZ-A 54825 and QCAZ-A 59067), *N. posadae* (QCAZ-A 57095), *N. humboldti* (QCAZ-A 71393, QCAZ-A 69108 and QCAZ-A 69110), *N. siren* (QCAZ-A 59075 and QCAZ-A 59069), *N. cochranae* (QCAZ-A 59074), and *N.* sp. (DHMECN 2249). Tissues were obtained from the genome bank of Museo de Zoología, Pontificia Universidad Católica del Ecuador (QCAZ), and the collection of Instituto Nacional de Biodiversidad (INABIO, DHMECN). DNA was extracted from liver or muscle tissue, preserved in 95% ethanol or tissue storage buffer, using guanidine thiocyanate protocol with some modifications. Primers used for amplification are listed in S1 Table.

PCR amplification was performed under standard protocols and amplicons were sequenced by the Macrogen Sequencing Team (Macrogen Inc., Seoul, Korea). Sequences were assembled using GeneiousPro 9.1.8 [51]. Vouchers and GenBank accession numbers for newly generated sequences are shown in Table 1.

We complemented our newly generated sequences with homologous sequences of *Nymphargus* from GenBank (http://www.ncbi.nlm.nih.gov/genbank) and representative species of other genera of Centrolenidae. The phylogeny was rooted with *Allophryne ruthveni*. Accession numbers for previously released GenBank sequences are available in the original publications [7,12,14,52,53], and [3]. We analyzed 10 genes: ND1, 12S, 16S (three mitochondrial genes), BDNF, CXCR4, C-MYC, POMC, RAG1, SLC8A1 and SLC8A3 (seven protein-coding genes). The alignment of the sequences was performed in GeneiousPro 9.1.8 [51] with the plug-in MAFFT [54] under default settings. Alignment errors were fixed manually with Mesquite v 3.70 software [55]. The final concatenated matrix had 102 individuals and 6613 bp and is available at https://zenodo.org/records/13323248 (https://doi.org/10.5281/zenodo.13323247).

We estimated phylogenetic relationships under maximum likelihood (ML) as optimality criterion. The matrix was partitioned by gene and codon position and each partition was analyzed under model GTR+R+I in software IQ-TREE 2.2.0.7 for Mac OS X [56,57]. To assess branch support, we ran 1000 ultra-fast bootstrap searches (-bb 1000 command) and 1000 replicates for the SH-like approximate likelihood ratio test (-alrt 1000 command) [58]. We assumed that branches with ultrafast bootstrap values > 94.9 and SH-aLRT values > 79.9 had strong support [59].

**Table 1. GenBank accession numbers for newly generated DNA sequences used in the phylogenetic analyses.**

| Species | Voucher | 12S | 16S | ND1 | C-MYC | RAG1 |
|---|---|---|---|---|---|---|
| *N. buenaventura* | DHMECN10902 | PQ559076 | – | MT733055 | – | PQ563246 |
| *N. buenaventura* | QCAZ54825 | – | MT734665 | MT733052 | – | – |
| *N. cochranae* | DHMECN 11616 | PQ559077 | – | PQ563238 | PQ563250 | PQ563247 |
| *N. dajomesae* sp. nov | QCAZ-A 68522 | PQ559078 | – | PQ563240 | PQ563252 | – |
| *N. dajomesae* sp. nov | QCAZ-A 68526 | PQ559080 | – | PQ563241 | PQ563253 | – |
| *N. dajomesae* sp. nov | QCAZ-A 68586 | PQ559079 | – | – | – | – |
| *N. humboldti* | QCAZ-A 69108 | – | PQ559072 | PQ563243 | – | – |
| *N. humboldti* | QCAZ-A 69110 | – | PQ559073 | PQ563244 | – | – |
| *N. humboldti* | QCAZ-A 71393 | – | PQ559074 | PQ563245 | – | – |
| *N.* aff. *pluvialis* | QCAZ-A 57095 | – | PQ559071 | PQ563242 | – | – |
| *N. siren* | DHMECN 11612 | – | – | PQ563239 | PQ563251 | PQ563249 |
| *N. siren* | DHMECN 11613 | PQ559075 | – | PQ563237 | – | PQ563248 |
| *Nymphargus* sp. | DHMECN 2249 | PX392541 | – | PX395926 | – | – |

To obtain a time-tree, we applied the least square dating method [60] as implemented in IQ-TREE using three secondary calibration points based on the time-tree published by [61]: 40.8 Mya for the root (divergence between *Allophryne ruthveni* and Centrolenidae), 23.1 My for the divergence between *Espadarana* and *Hyalinobatrachium* and 12.5 My for the divergence between *Espadarana* and *Teratohyla*.

### Nomenclatural acts

The electronic edition of this article conforms to the requirements of the amended International Code of Zoological Nomenclature, and hence the new names contained herein are available under that Code from the electronic edition of this article. This published work and the nomenclatural acts it contains have been registered in ZooBank, the online registration system for the ICZN. The ZooBank LSIDs (Life Science Identifiers) can be resolved and the associated information viewed through any standard web browser by appending the LSID to the prefix "http://zoobank.org/". The LSID for this publication is: urn:lsid:zoobank.org:pub:6B773209-CBB1–4DA8-B68E-E9C7B4420C1C. The electronic edition of this work was published in a journal with an ISSN and has been archived and is available from the following digital repositories: PubMed Central, and LOCKSS.

## Results

### Phylogeny

The time-calibrated ML tree (Fig 1) indicates that *Nymphargus* originated ~17 Mya, an age congruent with previous estimates [53]. As in previous studies (e.g., [3]), we found weak support for the most basal nodes within *Nymphargus* (Fig 1). Differences with previous phylogenies pertain to those and other weakly supported nodes. For example, *N. chancas* was sister to all *Nymphargus* in [3], whereas in our phylogeny, it falls within a clade with *N. balionotus*, *N. colomai*, *N. grandisonae*, *N. manduriacu*, and *N. mariae*. Only one clade older than 10 My is strongly supported. It shows *N. vicenteruedai* as sister to two well-supported clades (A and B in Fig 1).

Most species of *Nymphargus* originated during the Miocene-Pliocene. Exceptions include *N. posadae* [46], *N. ocellatus* [62], and *N. pluvialis* [63], which originated during the Pleistocene.

The new population discovered at Reserva Biológica El Quimi belongs to clade B, which includes two basally diverging subclades (Fig 1). One is distributed on the eastern flanks of the Andes of Ecuador (*N. lindae* + *N. cochranae*). Its sister clade is a composite of 10 species from both sides of the Andes, four of which are undescribed. Within the latter, the Reserva Biológica El Quimi population is sister to DHMECN 2249 (tissue QCAZ-A 59068), a putative unnamed species from San Juan Bosco, Morona Santiago province (previously misidentified as *N. cariticommatus* [64,65]; Fig 2). These lineages diverged from each other during the Pliocene, ~4.4 My.

According to our operational criteria, the population from Reserva Biológica El Quimi is a species because it has a deep genetic divergence and has diagnostic morphological differences characteristic of interspecific differences in *Nymphargus* (see Diagnosis section below). Moreover, according to our time-tree, it diverged from the closest named species, *N. griffithsi* and *N. lasgralarias*, ~6 Mya. Such a long divergence time is unlikely to separate populations of a single species. Our phylogeny based exclusively on nuclear genes (S1 Fig) shows that the new species is distinct from other *Nymphargus*, indicating that its recognition is not an artifact of mitonuclear discordance (see below). Based on the combined evidence, we conclude that the population from Reserva Biológica El Quimi represents a new species that we describe below.

Clade B also includes a subclade from the Western Cordillera of the Andes of Ecuador (*N. griffithsi* [15] + *N. lasgralarias* [24] + *N.* sp.). Specimens of the latter, KU 202801 and KU 202796, from Carchi, Ecuador, are sister to *N. lasgralarias*, similar to the result found by Guayasamin et al. [3]. However, we note that Guayasamin et al. [3] show the specimen KU 202801 twice in the same phylogeny, with the duplicate reported as "*N. garciae*" and sister to *N. vicenteruedai*. They also refer to specimen KU 202796 both as "*N.* aff. *griffithsi*" and "*N. garciae*" in the text.

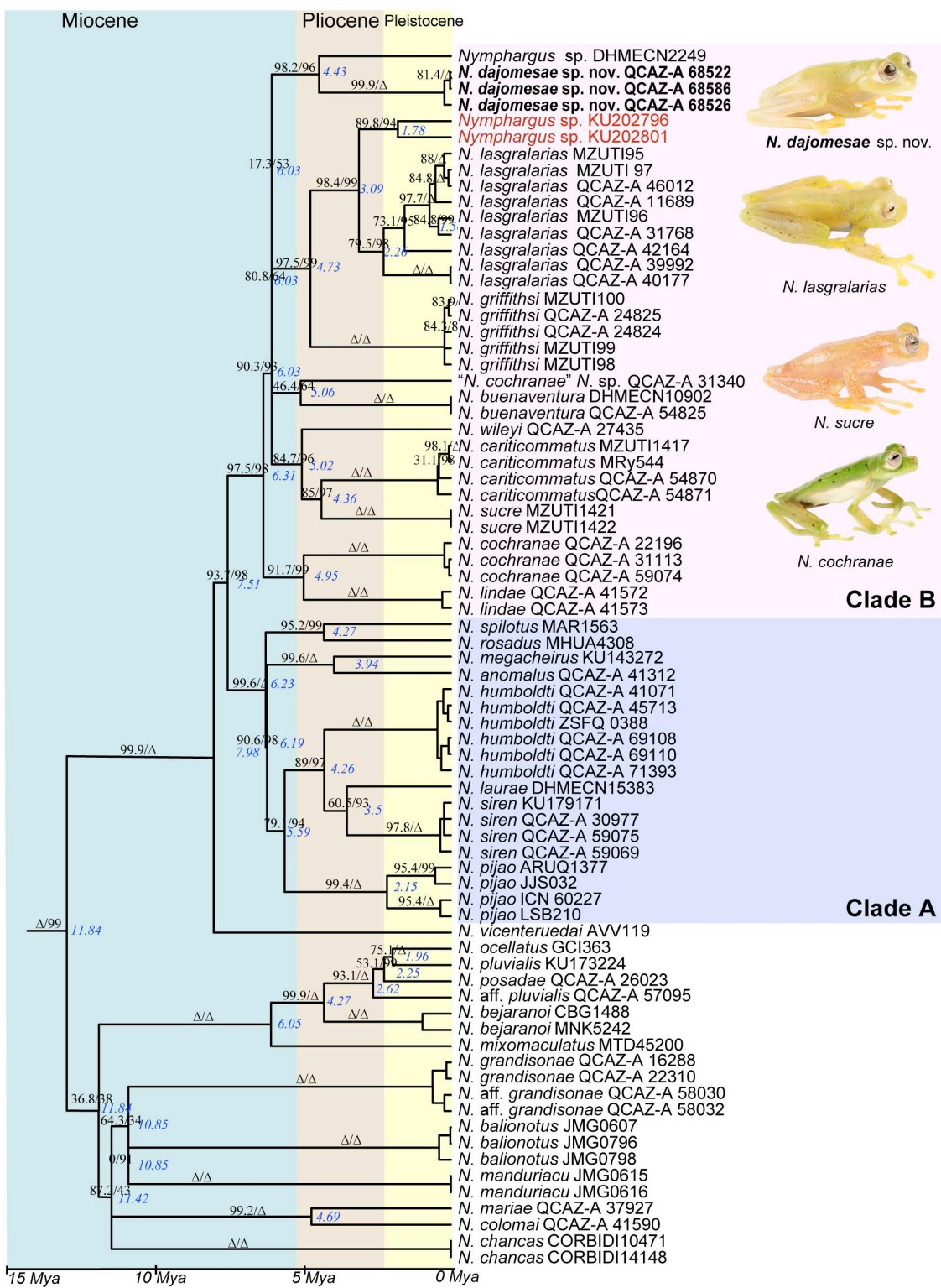

**Fig 1. Maximum likelihood tree of *Nymphargus* inferred from a partitioned analysis of 6613 aligned sites of DNA sequences of 10 nuclear and mitochondrial loci.** SH-aLRT support (before slash) and ultra-fast bootstrap support (after) are shown as percentages on branches; triangles indicate values of 100. Node ages, in My, are shown in blue (ages lower than 1.0 Mya not shown). The species name is followed by the number of the specimen

voucher. Genera other than *Nymphargus* are not shown. The new species is highlighted in bold. Each specimen in red was referred both as "*N.* aff. *griffithsi*" and "*N. garciae*" in Guayasamin et al. [3]. Specimens with photographs: *Nymphargus dajomesae* sp. nov QCAZ-A 68522 (photo BIOWEB-Museo QCAZ archive); *Nymphargus lasgralarias*, QCAZ-A 40177 (photo S. R. Ron); *Nymphargus sucre*, QCAZ-A 68099; and *Nymphargus cochranae* QCAZ-A 59370 (photo Juan Carlos Sánchez, BIOWEB-Museo QCAZ archive).

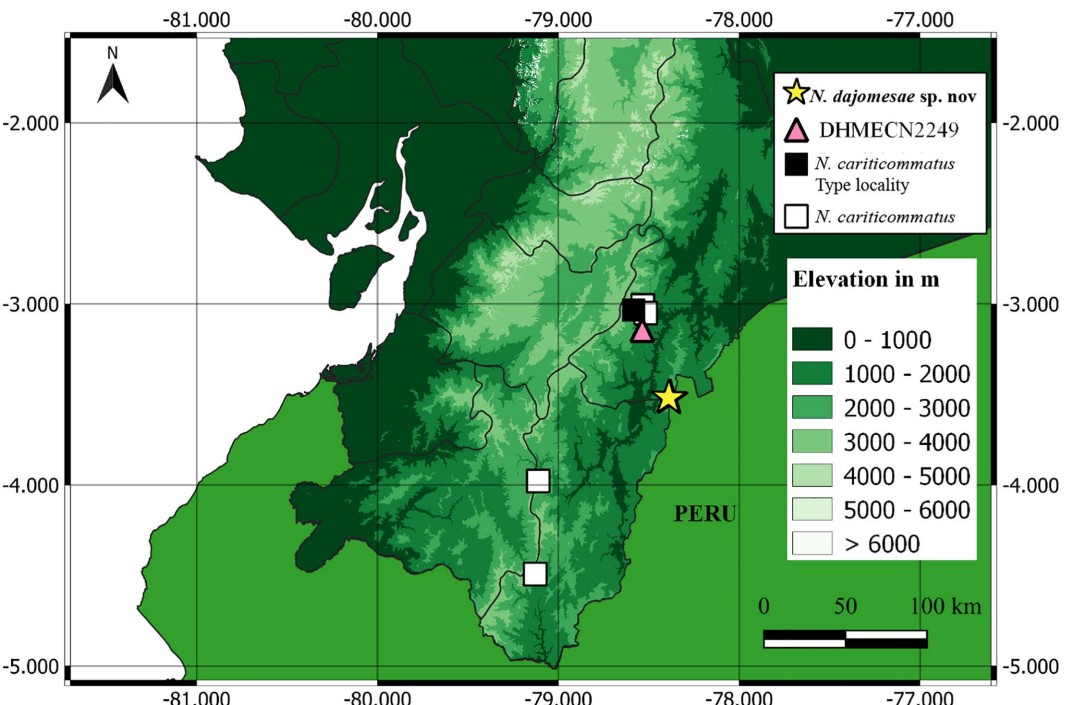

**Fig 2. Distribution of *Nymphargus dajomesae* sp. nov. in Ecuador.** Type locality of *N. cariticommatus* KU 202803 [19]; *N. cariticommatus* MZUTI 1417, USNM 288435, DHMECN 1974, QCAZ-A 33977; *N.* sp DHMECN 2249. Geographical boundaries obtained from Natural Earth (https://www.naturalearthdata.com/) and elevation raster from USGS Earth Resources Observatory and Science (EROS) Center (https://www.usgs.gov/centers/eros).

Similarly to the concatenated tree (Fig 1), in the phylogeny based on nuclear genes (S1 Fig), the El Quimi population is monophyletic and closely related to *N. lasgralarias*, *N. wileyi*, *N.* sp. DHMECN 2249, *N.* sp. QCAZ-A 31340, and *N. buenaventura*. However, as expected for nuclear genes, node support is generally weak. Unlike the concatenated phylogeny, the El Quimi population is sister to *N. lasgralarias*, a species found on the opposite side of the Andes.

## Systematic account

### *Nymphargus dajomesae* sp. nov.

urn:lsid:zoobank.org:act:ECC9AD28-F16A-4B58-8E13-CD6DA3A82A9E

**Proposed Spanish common name:** Rana de cristal de Dajomes
**Proposed English common name:** Dajomes Glassfrog

## Holotype

Fig 3. 5. QCAZ-A 68586, adult male; buffer zone Reserva Biológica El Quimi (3.5192° S, 78.3837° W; 2070 m), provincia de Morona Santiago, República del Ecuador, collected by Diego Almeida, Darwin Núñez, Eloy Nusirquia, Alex Achig, and Ricardo Gavilanes on 07 July 2017.

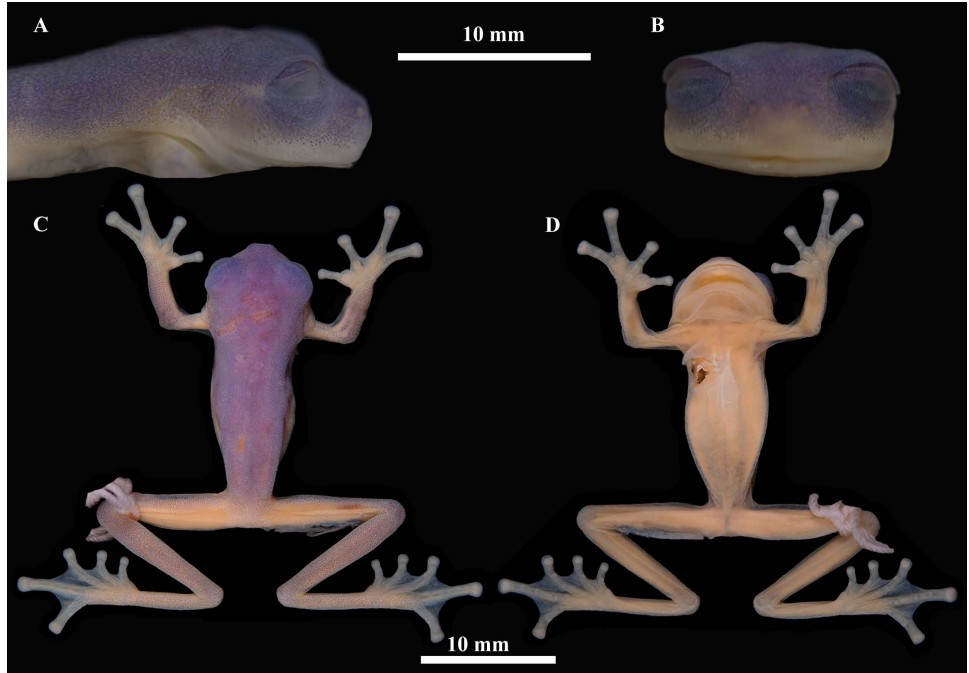

**Fig 3. Preserved holotype of *Nymphargus dajomesae* sp. nov. QCAZ-A 68586, adult male, SVL = 23.3 mm. Reserva Biológica El Quimi, Morona Santiago Province, Ecuador. (A) lateral view of head, (B) frontal view of head, (C) dorsal view, (D) ventral view. Photographs by Mylena Masache.**

## Paratypes

6 adult males, QCAZ-A 68522–26, same collection data as holotype. QCAZ-A 72521, buffer zone Reserva Biológica El Quimi (3.51926°S, 78.37752°W, 2111 m), provincia de Morona Santiago, República del Ecuador, Diego Almeida, Darwin Núñez, Eloy Nusirquia, Alex Achig, and María del Mar Moretta on 13 April 2018.

## Etymology

The specific epithet is a noun in the genitive case honoring Neisi Dajomes, the first Ecuadorian woman to win a gold medal at the Olympic Games (Tokyo 2020, women's 76 kg weightlifting). In addition, she has won gold medals at the World and Pan American Weightlifting Championships, and the Pan American, Bolivarian and South American games.

## Definition

*Nymphargus dajomesae* differs from all other Centrolenidae by the following combination of characters: (1) dentigerous process of vomer and vomerine teeth absent; (2) snout truncated in dorsal and lateral views; loreal region slightly concave; (3) tympanic annulus, lower ¾ visible, upper border covered by supratympanic fold; tympanic membrane colored as surrounding skin; (4) dorsal skin shagreen with microspicules; (5) ventral skin granular; pericloacal area granular, enameled; pair of sub-cloacal warts present; (6) parietal peritoneum white, iridophores covering almost entirely (condition P4); peritonea covering heart, esophagus, stomach and kidneys covered by iridophores (V2); all other peritonea clear; (7) liver lobed and hepatic peritoneum clear (condition H0); (8) humeral spines absent in adult males; (9) webbing absent between fingers I and II, basal between fingers II and III; reduced between outer fingers, webbing formula III ($3^-$ –$2^{1/2}$) IV; (10) foot webbing formula: I ($2^-$)–($2^-$–$2^+$) II (1– $1^{1/2}$)–($2^+$–$2^{1/2}$) III ($1+$–$1^{2/3}$)–($2$–$2^+$) IV ($2^{1/2}$–$2^{2/3}$)–($1^{2/3}$–$2^-$) V; (11) ulnar folds low

or absent, no tubercles on hands or feet; fold along the external edge of hand and finger IV. Tarsal fold low, smooth, and without tubercles; (12) nuptial pad Type I; prepollex concealed; (13) Finger II slightly longer than Finger I; (14) ED larger than Fin3DW; (15) in life, dorsum yellowish green to dark green, without pale or dark marks; fingers and toes yellow; bones green; (16) in preservative, dorsum lavender without pale or dark marks, fingers and toes cream; (17) in life, iris tan to greyish tan with fine dark reticulations and punctuations, more conspicuous on upper half; and pale yellow circum-pupillary ring present; (18) melanophores reaching Finger IV and toes IV and V; (19) call consists of one click repeated at a rate of 0.56 calls/sec, dominant frequency range 4142.6–4166.0 Hz; (20) fighting behavior unknown; (21) egg clutches and parental care unknown; (22) tadpoles unknown; (23) medium body size; SVL in adult males 20.8–27.2 mm (22.4 ± 2.2 mm; n = 7), females unknown.

**Diagnosis**

*Nymphargus dajomesae* is diagnosed from all other species of Centrolenidae, except *N. chami* [46], *N. cristinae* [46], *N. griffithsi* [26], *N. lasgralarias* [26], *N. prasinus* [18], *N. posadae* [46], and *N. wileyi* [47], by having a combination of dorsum uniformly green without pale or dark marks, fingers and toes yellow, pericardium white (covered by irido-phores), medium body size, reduced hand webbing and absence of humeral spines in males. *Nymphargus chami* differs from *N. dajomesae* (characters of *N. dajomesae* in parentheses) by having vomerine teeth (absent), green dorsal with abundant subconical tubercles (shagreen), dorsum green with pale tubercles (uniformly green), esophageal and stomach peritonea translucent (white) and larger body size (SVL 30.5–34.6 mm in males of *N. chami* vs. 20.8–27.2 mm in males of *N. dajomesae*). *Nymphargus cristinae* has dorsum green with occasional scattered black flecks (uniformly green), dorsal surfaces of legs with black flecks (absent), fingers and toes pale green (yellow), esophageal and stomach peritonea translucent (white), and slightly larger body size (SVL 26.0–31.1 mm in males of *N. cristinae* vs. 20.8–27.2 mm in males of *N. dajomesae*). *Nymphargus griffithsi* and *N. lasgralarias* differ by having esophageal and stomach peritonea translucent (white), fingers and toes green (yellow), iris golden yellow or whitish cream with numerous dark punctuations (tan to greyish tan with fine dark reticulations and punctuations), and some individuals of *N. griffithsi* have dark flecks (absent). *Nymphargus prasinus* differs by its dorsal skin smooth to finely shagreen (shagreen), fingers and toes pale green (yellow), snout round in dorsal view (truncate), esophageal and stomach peritonea translucent (white), and larger body size (SVL 33.6–34.5 mm in males of *N. prasinus* vs. 20.8–27.2 mm in *N. dajomesae*). *Nymphargus posadae* differs by its dorsal skin with numerous small warts (shagreen), ventrolateral border of arm, Finger IV, tarsus, and Toe V white (barely perceptible white in *N. dajomesae*), esophageal and stomach peritonea translucent (white), and larger body size (SVL 30.7–34.1 mm in males of *N. posadae* vs. 20.8–27.2 mm in males of *N. dajomesae*). *Nymphargus wileyi* has esophageal and stomach peritonea translucent (white), fingers and toes green (yellow), and coppery white iris with black reticulations (tan to greyish tan with fine dark reticulations and punctuations). The only other species of glassfrog similar to *N. dajomesae* is *N. cariticommatus*, however, the latter has snout rounded in dorsal view and bluntly rounded in lateral view (truncate), dorsum green with small pale flecks (uniformly green) and stomach peritoneum translucent (white).

Other species of *Nymphargus* from the Cordillera del Condor and the eastern Andean slopes of southern Ecuador and northern Peru differ as follows from *Nymphargus dajomesae* (characters in parentheses): *Nymphargus anomalus* has dorsal skin shagreen with elevated warts (shagreen without warts), esophageal and stomach peritonea translucent (white), dorsum tan brown with dark flecks and ocelli black with orange centers (uniformly green). *Nymphargus cochranae* and *N. lindae* have esophageal and stomach peritonea translucent (white) and dorsum green with ocelli black with yellow to orange center (uniformly green). *Nymphargus chancas* and *N. colomai* have dorsum tan, green or brown with pale flecks or dots (uniformly green), iris white with horizontal stripe dark (tan to greyish tan with fine dark reticulations and punctuations). *Nymphargus mariae* has vomerine teeth (absent), dorsum shagreen with flat warts (shagreen), esophageal and stomach peritonea translucent (white), dorsum green with yellow spots or reticulation (uniformly green), and larger body

size (SVL 25.4–31.7 mm in males of *N. mariae* vs. 20.8–27.2 mm in males of *N. dajomesae*). *Nymphargus sucre* has esophageal and stomach peritonea translucent (white) and dorsum brownish yellow with yellow dots (uniformly green).

## Description of holotype

Adult male, medium-sized and slender body, SVL = 21.3 mm. Head distinct, slightly wider than long (HL/HW = 0.76, HW/SVL = 0.36, HL/SVL = 0.27. Snout short (EN/HL = 0.33); rounded in dorsal view and truncate in profile; nostril protuberant, closer to tip of snout than to eye, directed front-laterally, producing a shallow depression in internarial area; canthus rostralis indistinct, slightly concave loreal region; lips slightly flared; (Fig 3). Eyes large (ED/HL = 0.45), directed anterolaterally at ~40º from midline; interorbital area wider than eye diameter (IOD/ED = 1.31, EN/ED = 0.74, EN/IOD = 0.57). Tympanic annulus oriented vertically with slight posterolateral inclination; weak supratympanic fold from behind eye to insertion of arm; tympanic membrane pigmented like surrounding skin. Dentigerous processes of vomer absent; choanae large, round; tongue ovoid, unnotched; vocal slits paired, extending from mid-tongue to angles of jaws.

Skin of dorsal surfaces of head, body, and flanks of head and body shagreen. Skin of ventral surfaces granular (Fig 4). Cloacal opening directed posteriorly to the upper level of thighs; pericloacal area granular and enamelled posterodorsally; two enlarged, round, flat subcloacal warts on ventral surfaces of thighs below vent.

Upper arm thin, forearm moderately robust, width of upper arm about half that of forearm. Humeral spines absent. No tubercles on hands, forearms, or feet; fold along the external border of the hand and finger IV. Relative lengths of fingers: III > IV > II > I; webbing absent between fingers I and II, basal webbing between fingers II and III, outer fingers webbing formula III ($^{2/3}$–2$^{1/2}$) IV; bulla absent. Finger discs wider than adjacent phalanx and slightly truncated, disc of Finger III slightly larger than those on toes, and smaller than eye diameter (ED/Fin3DW = 0.55). Subarticular tubercles rounded and elevated, supernumerary tubercles present; palmar tubercle large, rounded, and elevated; thenar tubercle elliptic. Prepollex concealed, nuptial excrescences Type I, ovoid, granular, and unpigmented, extending from ventrolateral base to dorsal surface of Finger I, covering its base.

Hind limbs slender (TL/SVL = 0.54, FL/SVL = 0.39). Outer tarsal fold low, smooth and without tubercles. Inner metatarsal tubercle large, ovoid; outer metatarsal tubercle not evident. Subarticular tubercles small, round; supernumerary tubercles

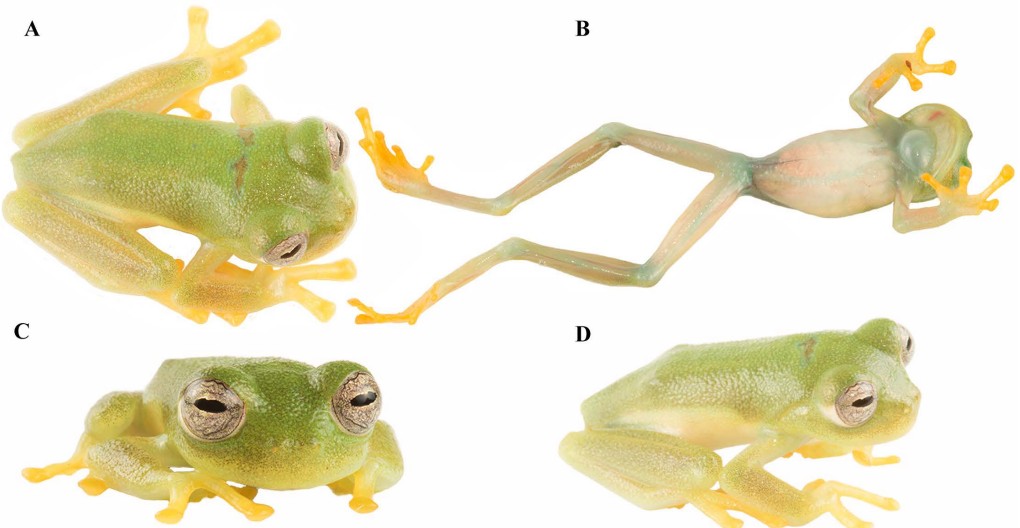

**Fig 4. Live holotype of *Nymphargus dajomesae* sp. nov. QCAZ-A 68586.** (A) dorsal view, (B) ventral view, (C) Frontal view and (D) lateral view. Photos by BIOWEB-Museo QCAZ-A archive.

low, indistinct. Foot webbing formula: I2⁻–2⁺II1⁺–2⁺III1$^{1/3}$–2⁺IV2$^{1/2}$–1$^{2/3}$V; toe discs and disc pads rounded, papillae in tip of discs of toes absent (Fig 5).

**Coloration of holotype in life.** (Fig 4) Dorsal surfaces of head and body bright green, those of fore and hindlimbs yellowish green; flanks changing from bright green to pale green; ventral surfaces whitish cream; fingers and toes yellow in dorsal and ventral views; lips pale green. Iris tan with dark grey fine reticulations and punctuations, sclera white. Bones green.

**Coloration of holotype in preservative.** (Fig 3) Dorsal surfaces of head and body lavender, those of fore and hindlimbs paler lavender; lip, fingers, toes and ventral surfaces cream. Parietal peritoneum white, covered almost entirely by iridophores; pericardium and esophageal, stomach and renal peritonea white (covered by iridophores), renal peritoneum with small translucent flecks; all other peritonea clear.

**Measurements of holotype in mm.** SVL, 21.3; HW 7.61; HL, 5.76; ED, 2.58; IOD, 3.38; EN, 1.91; IND, 1.89; TL, 11.52; FL, 8.4; Fin3DW, 1.42.

## Variation

Variation in color of live and preserved individuals is shown in Figs 6 and 7. Iris background coloration varies from tan to greyish tan, with dark reticulations and punctuations that, in some specimens, become denser (e.g., QCAZ-A 68522). Paratypes QCAZ-A 68523, 68524 have subcloacal warts more rounded than the holotype. QCAZ-A 68523 and QCAZ-A

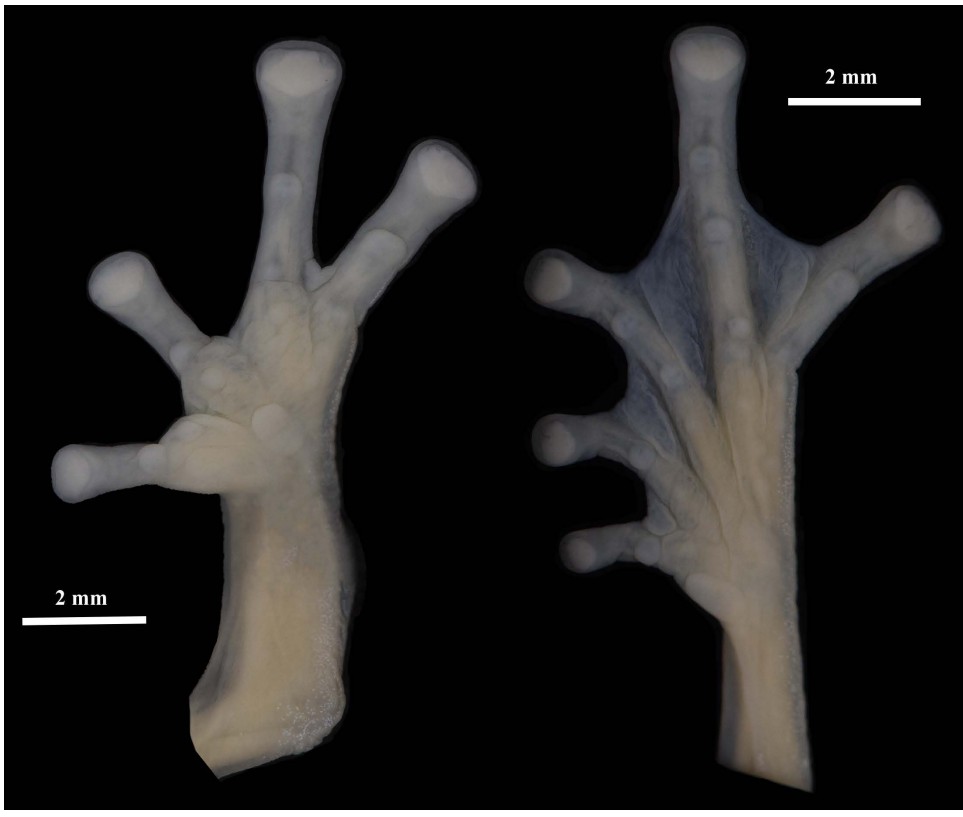

**Fig 5. Ventral views of the right hand and foot of *Nymphargus dajomesae* sp. nov. Holotype (QCAZ-A 68586). Photos by Mylena Masache.**

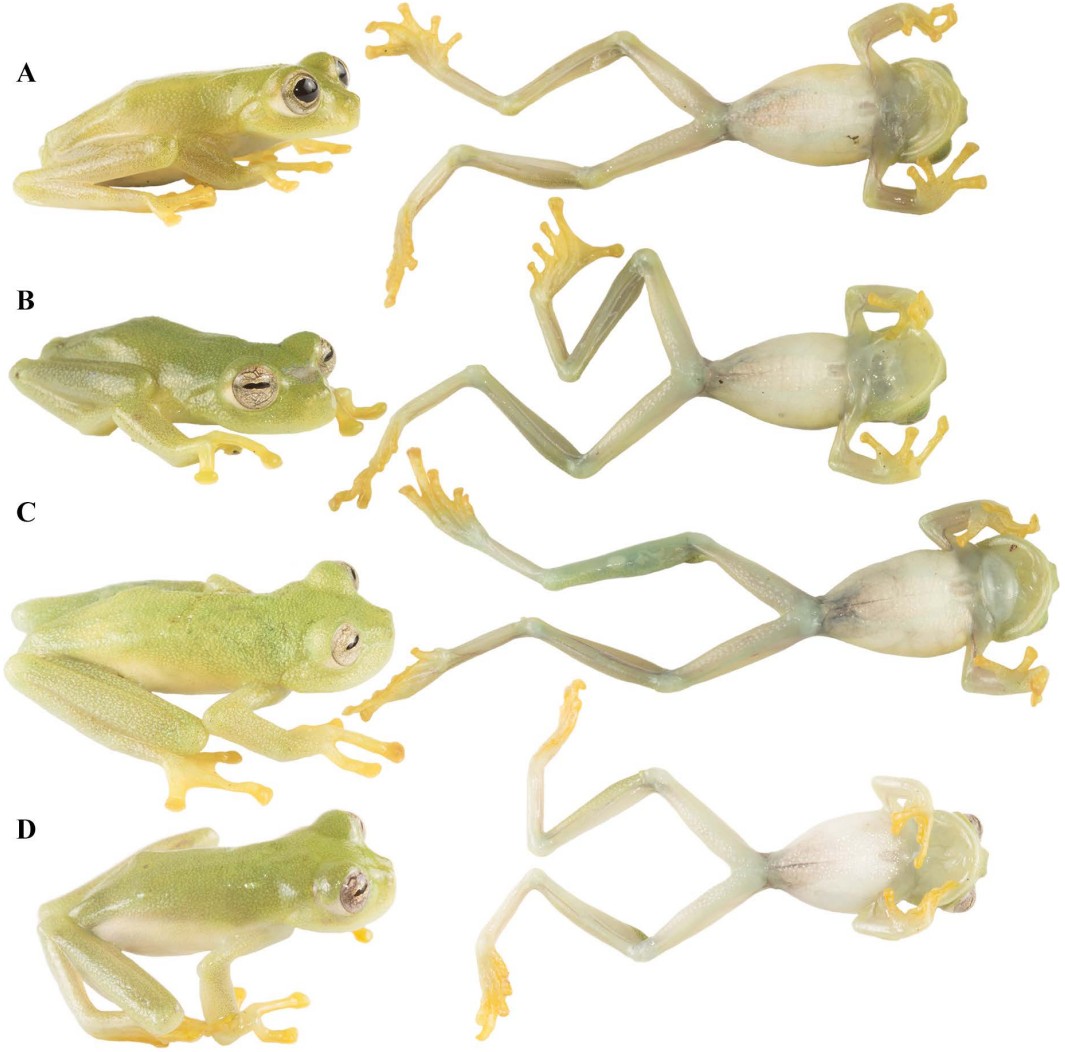

**Fig 6. Variation of *Nymphargus dajomesae* sp. nov. in life.** Dorsal and ventral view of some paratypes, all adult males: **(A)** QCAZ-A 68522 (SVL 22.4), **(B)** QCAZ-A 68524 (SVL 22.0), **(C)** QCAZ-A 68526 (SVL 27.2), **(D)** QCAZ-A 72521 (SVL 20.8). Photos by BIOWEB-Museo QCAZ-A archive.

68525 have darker skin on the arms and legs than the holotype. QCAZ-A 68524 presents a low ulnar fold, absent in the holotype; QCAZ-A 68526 has lighter skin. Morphometric variation is shown in Table 2.

## Advertisement call

Description based on a male QCAZ-A 68524, recorded at the type locality on July 8, 2017, at 20:30 h. The advertisement call is a single pulsed note with a click like sound. The call is repeated at a rate of 33.6 calls/minute (Fig 8). Its average call duration is 0.049 seconds (SD = 0.003 s, range 0.0453–0.052), average dominant frequency is 4154.3 Hz (SD = 11 Hz, range 4142.6–4166.0) with a 90% bandwidth = 243.7 Hz (SD = 15.27, range 222–246). Up to four harmonics are visible in the spectrogram; the dominant frequency appears to be the fundamental as the second harmonic has ~2 times the frequency of the first (Fig 8E). The peak time occurs at 0.0105 seconds (SD = 0.0031 s, range 0.0061–0.0139). Calls of the sister species of *N. dajomesae* are unknown.

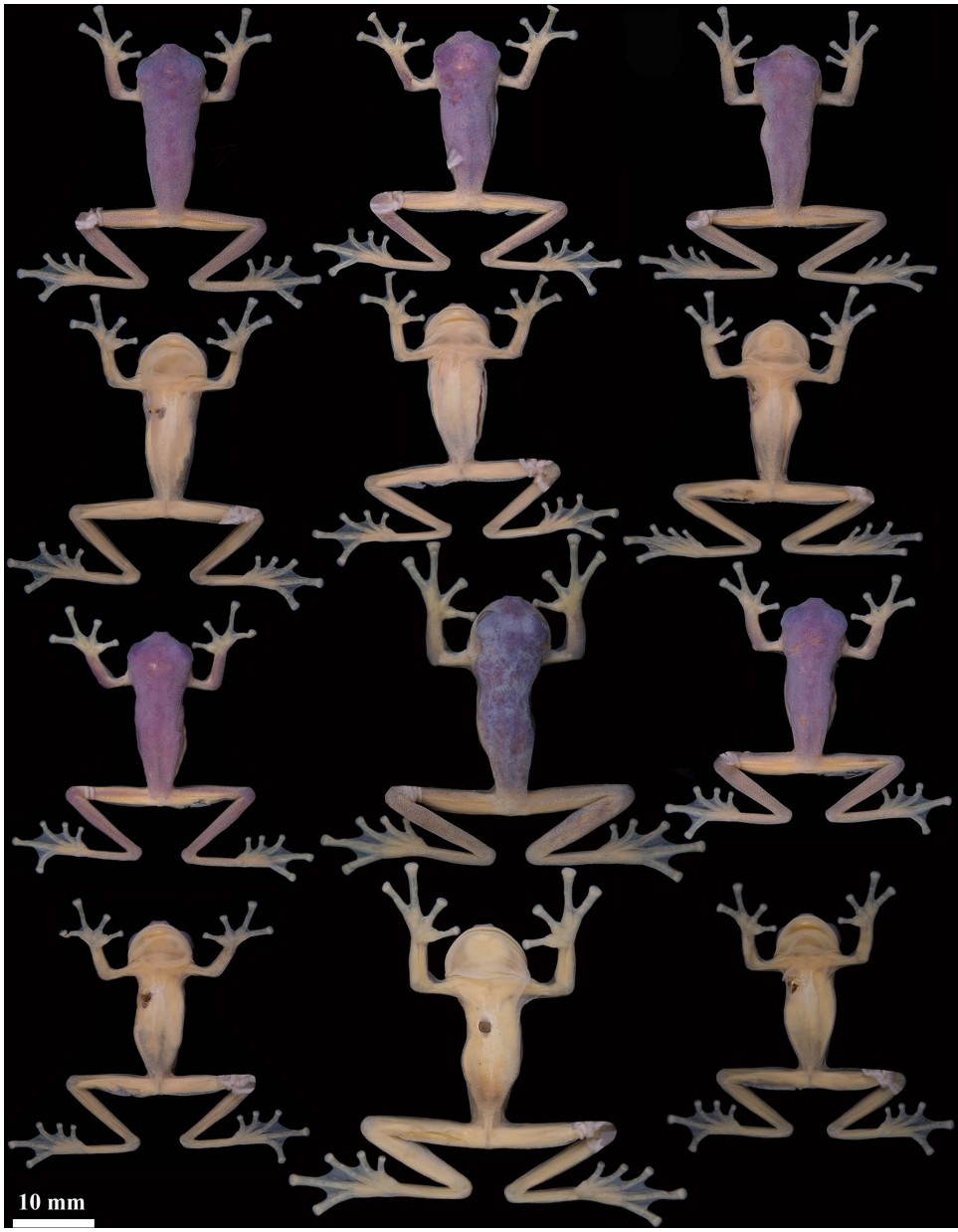

**Fig 7. Variation of preserved specimens of *Nymphargus dajomesae* sp. nov.** Dorsal and ventral view from left to right, first and second rows: QCAZ-A 68522, 68523, 68524; third and fourth rows: QCAZ-A 68525, 68526, 68586; All adult males. Photographs by Mylena Masache.

### Distribution, ecology, conservation status, and extinction risk

*Nymphargus dajomesae* is known from a single locality at Reserva Biológica El Quimi, Morona Santiago province, Ecuador, on Cordillera del Cóndor between 1992–2090 m (Fig 2). Specimens were collected at the second plateau on the eastern side of the valley of Quimi. This plateau is made up of quartzite sandstone and is considered the geological base of Cordillera del Cóndor and the highest plateau of the Hollin Formation (2000 m–2200 m) [25,66]. The type locality is within Eastern Montane Forest (sensu [25]) or Evergreen Montane Forest on sandstone plateaus of Cordillera del Cóndor

**Table 2. Morphometric measurements (in mm) of *Nymphargus dajomesae* sp. nov. All individuals are adult males. Abbreviations: Snout–vent length (SVL), head width (HW), head length (HL), horizontal eye diameter (ED), inter–orbital distance (IOD), eye-nostril distance (EN), internarial distance between nostrils (IND), Finger III disc width (Fin3DW), tibia length (TL), foot length (FL), tympanum diameter (TD).**

| Measurement | Range | Average±SD |
|---|---|---|
| SVL | 20.8–27.2 | 22.4±2.2 |
| HW | 7.2–10.2 | 8.1±1.0 |
| HL | 5.7–8.3 | 6.3±1.0 |
| ED | 2.6–2.9 | 2.8±0.1 |
| IOD | 3.4–4.2 | 3.6±0.3 |
| EN | 1.4–2.3 | 1.8±0.3 |
| IND | 1.8–2.4 | 1.9±0.2 |
| TL | 11.5–15.6 | 12.5±1.5 |
| FL | 8.4–12.8 | 9.4±1.6 |
| Fin3DW | 1.1–2.1 | 1.5±0.3 |

(sensu [67]). Forests at sandstone plateaus in the region are characterized by low and open canopy with scattered trees 10–15 m high, shrubby vegetation, and soils covered by mosses, roots, and terrestrial bromeliads (this type of soil cover is locally known as *bamba*) (Fig 9). Most individuals were active at night, found perched on leaves between 60–180 cm above ground, near streams. QCAZ-A 68523 was found on bromeliads; QCAZ-A 68522, QCAZ-A 68524, QCAZ-A 68586 were calling.

Based on IUCN Red List criteria [68], *N. dajomesae* should be considered as Data Deficient (DD). Surveys of amphibians in Cordillera del Cóndor have been scant and sporadic [25]. The population status of *N. dajomesae* is unknown, and the possibility of the discovery of additional populations cannot be ruled out. The type locality is 5 km from agricultural areas [67] and 7 km from a large-scale mining exploitation. Mining in Cordillera del Cóndor has negatively impacted amphibian populations [41] and could threaten *N. dajomesae* in the future.

## Discussion

We described a new species of *Nymphargus* from one of the sandstone table-top plateaus of Cordillera del Cóndor. *Nymphargus dajomesae* is the fourth species of *Nymphargus* known from Cordillera del Cóndor. The other three are *N. cochranae*, *N. colomai*, and *N. lindae* [3]. Only one of them, *N. cochranae*, is not endemic to Cordillera del Cóndor. In our phylogeny, *N. cochranae* was polyphyletic because one specimen from southern Ecuador, QCAZ-A 31340, was not closely related to *N. cochranae* from central Ecuador (QCAZ-A 22196, 31113, 59074) indicating that more than one species is hidden under that binomen. Because the type locality of *N. cochranae* is closer to the clade from central Ecuador, we tentatively consider the southern population as an undescribed species. The polyphyly of "*N. cochranae*" provides a remarkable example of color convergence in Centrolenidae.

*Nymphargus dajomesae* is sister to a candidate species from the eastern slopes of Cordillera Oriental. Although the airline distance between them is only 45 km, both species are separated by the lowland valley of the Zamora River (800–650 m of elevation) which could represent a geographic barrier to gene flow mediating allopatric speciation. Additional inventories in the region are necessary to determine their distribution ranges to test the putative role of environmental barriers in their speciation.

The first expeditions to the Cordillera del Cóndor failed to find glassfrogs [23,69,70]. However, subsequent surveys found several new species of centrolenids, most of them being endemic to the area [3,11,21,71–73]. At the plateaus of El Quimi reserve, two field trips in 2017 and 2018, with a duration of 22 days in total, have resulted, so far, in the description

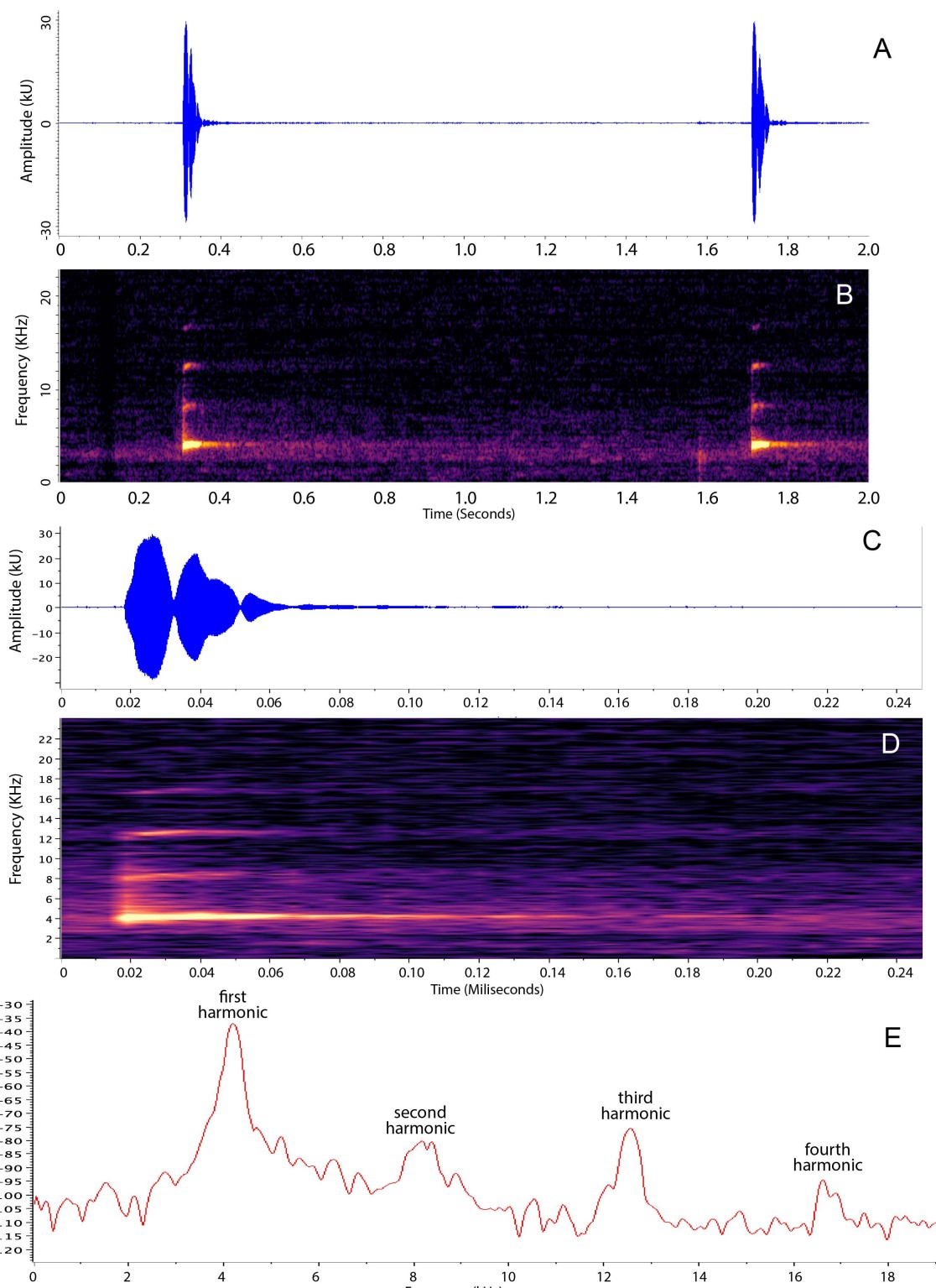

**Fig. 8. Advertisement call of *Nymphargus dajomesae* sp. nov.** Panels A and B show two consecutive calls; panels C, D, and E show a detailed view to the first of those calls. A and C are oscillograms; B and D spectrograms; E power spectra. Call from male QCAZ-A 68524 from Reserva Biológica El Quimi (3.5192º S, 78.3837º W; 2070 **m)**, Morona Santiago province, Republic of Ecuador. Recorded by Diego Almeida.

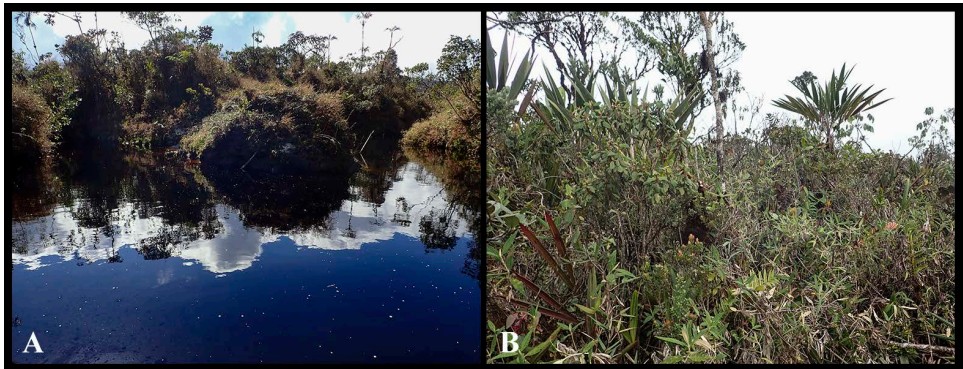

**Fig. 9. Habitat at the type locality, Reserva Biológica El Quimi. (A)** Slow-flowing blackwater stream rich in tannins. **(B)** Surrounding vegetation composed by dense shrubs, bromeliads, and mosses.

of two species of *Pristimantis* [39], one *Hyloscirtus* [25], one *Pholidobolus* [74], and two plants [75,76]. Moreover, six new species of *Pristimantis*, one *Hyloxalus*, and one *Bolitoglossa* are waiting to be described [77]. The only described species found were *Centrolene condor* and *Excidobates condor*. Therefore, over 85% of the species of the amphibian community at El Quimi plateau were new.

These findings highlight the need for continuing biodiversity surveys and taxonomic efforts in the Subandean cordilleras of southeastern Ecuador and northeastern Peru (i.e., Kutukú, Condor, Kampankis) where some ecosystems are "lost worlds" of amphibian diversity.

Communities with large proportions of unnamed species, however, seem to be restricted to the highlands (> 1800 m). At El Quimi, the amphibian community found below the plateau, on the adjacent valley of El Quimi river (< 1000 m above sea level), was composed entirely by described species: *B. almendarizae, B. cinerascens, B. lanciformis, Dendropsophus rhodopeplus, Leptodactylus wagneri, Lithodytes lineatus*, and *Rhinella poeppigii* [77]. Therefore, future inventories at Cordillera del Cóndor should invest more sampling effort on montane habitats.

Our results support Cisneros-Heredia & Yánez-Muñoz's [9] hypothesis of a clade formed by *N. buenaventura, N. cariticommatus, N. griffithsi* [15], and *N. wileyi* [47], with the addition of four undescribed species at the time: "*N.* aff. *cochranae*" [3,53] (QCAZ-A 31340), *N. dajomesae, N. lasgralarias*, and *N. sucre*. Interestingly, *N. buenaventura*, is not sister to the most morphologically similar and geographically closer *N. griffithsi*. This arrangement is unlikely to change as support for the clade joining *N. griffithsi* + *N. lasgralarias* + *N.* sp. is strong. However, the close relationship between *N. buenaventura* and *Nymphargus* sp. (QCAZ-A 31340) is low. Therefore, the closest relative of *N. buenaventura* has yet to be confirmed. Remarkably, *N. buenaventura* and *N.* aff. *cochranae* are among the oldest species of *Nymphargus*, with a divergence estimate of ~5.0 My.

## Supporting information

**S1 Table. Primers used in this study.**
(PDF)

**S1 Fig. Maximum likelihood tree of *Nymphargus* inferred from a partitioned analysis of 3536 aligned sites of DNA sequences of 7 nuclear loci.** SH-aLRT support (before slash) and ultra-fast bootstrap support (after) are shown as percentages on branches. The species name is followed by the number of the specimen voucher. Genera other than *Nymphargus* are not shown.
(TIF)

**S1 Appendix. Specimens examined.**
(PDF)

## Acknowledgments

We thank Alex Achig, Diego Almeida, Gabriela Galarza, Ricardo Gavilanes, María del Mar Moretta, Darwin Núñez, Eloy Nusirquia, Daniela Pareja, Diego Paucar for the field work and for collecting the specimens and audio recordings used in this study. Mario Yanez-Muñoz provided tissues used in this study. Field and laboratory work in Ecuador was funded by Secretaría Nacional de Educación Superior, Ciencia, Tecnología e Innovación del Ecuador SENESCYT (Arca de Noé initiative; SRR and Omar Torres principal investigators) and grants from Pontificia Universidad Católica del Ecuador, Dirección General Académica.

## Author contributions

**Conceptualization:** Diego F. Cisneros-Heredia, Santiago R. Ron.

**Data curation:** Mylena V. Masache-Sarango, Diego F. Cisneros-Heredia, Santiago R. Ron.

**Formal analysis:** Mylena V. Masache-Sarango, Diego F. Cisneros-Heredia, Santiago R. Ron.

**Funding acquisition:** Santiago R. Ron.

**Investigation:** Mylena V. Masache-Sarango, Diego F. Cisneros-Heredia, Santiago R. Ron.

**Methodology:** Mylena V. Masache-Sarango, Diego F. Cisneros-Heredia.

**Project administration:** Santiago R. Ron.

**Supervision:** Diego F. Cisneros-Heredia, Santiago R. Ron.

**Validation:** Mylena V. Masache-Sarango, Diego F. Cisneros-Heredia.

**Visualization:** Mylena V. Masache-Sarango, Diego F. Cisneros-Heredia.

**Writing – original draft:** Mylena V. Masache-Sarango, Diego F. Cisneros-Heredia, Santiago R. Ron.

**Writing – review & editing:** Mylena V. Masache-Sarango, Diego F. Cisneros-Heredia, Santiago R. Ron.

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
