## [Decision Letter · Decision Letter 0]

16 Oct 2025

Dear Dr. Ron,

We look forward to receiving your revised manuscript.

Kind regards,

Philippe J. R. Kok, Ph.D.

Academic Editor

PLOS ONE

Journal Requirements:

3. We note that Figure 2 in your submission contain [map/satellite] images which may be copyrighted. All PLOS content is published under the Creative Commons Attribution License (CC BY 4.0), which means that the manuscript, images, and Supporting Information files will be freely available online, and any third party is permitted to access, download, copy, distribute, and use these materials in any way, even commercially, with proper attribution. For these reasons, we cannot publish previously copyrighted maps or satellite images created using proprietary data, such as Google software (Google Maps, Street View, and Earth). For more information, see our copyright guidelines: http://journals.plos.org/plosone/s/licenses-and-copyright.

4. Please take this opportunity to be sure you have met all of our guidelines for new species. For proper registration of a new zoological taxon, we require two specific statements to be included in your manuscript.

a.        In the Results section, the globally unique identifier (GUID), currently in the form of a Life Science Identifier (LSID), should be listed under the new species name, for example:

Anochetus boltoni Fisher sp. nov. urn:lsid:zoobank.org:act:B6C072CF-1CA6-40C7-8396-534E91EF7FBB

Another LSID for the manuscript itself should also appear within the Nomenclature statement. You will need to contact Zoobank (zoobank.org/About) to obtain a GUID (LSID). You should receive one LSID for your manuscript and a separate, unique LSID for the new species.

b.        Please also insert the following text into the Methods section, in a sub-section to be called "Nomenclatural Acts":

The electronic edition of this article conforms to the requirements of the amended International Code of Zoological Nomenclature, and hence the new names contained herein are available under that Code from the electronic edition of this article. This published work and the nomenclatural acts it contains have been registered in ZooBank, the online registration system for the ICZN. The ZooBank LSIDs (Life Science Identifiers) can be resolved and the associated information viewed through any standard web browser by appending the LSID to the prefix "http://zoobank.org/". The LSID for this publication is: urn:lsid:zoobank.org:pub: XXXXXXX. The electronic edition of this work was published in a journal with an ISSN, and has been archived and is available from the following digital repositories: PubMed Central, LOCKSS [author to insert any additional repositories].

All PLOS ONE articles are deposited in PubMed Central and LOCKSS. If your institute, or those of your co-authors, has its own repository, we recommend that you also deposit the published online article there and include the name in your article.

Following a recent ruling by the International Commission on Zoological Nomenclature, electronic journals are now a valid format for publication of new zoological taxa. In order to ensure the valid publication of your new species, please be sure to include the updated version of Nomenclatural Acts (above). A complete explanation of our guidelines for publishing new species can be found on our website: http://www.plosone.org/static/guidelines#zoological.

5. We note that you have indicated that there are restrictions to data sharing for this study. PLOS only allows data to be available upon request if there are legal or ethical restrictions on sharing data publicly. For more information on unacceptable data access restrictions, please see http://journals.plos.org/plosone/s/data-availability#loc-unacceptable-data-access-restrictions.

Additional Editor Comments:

Reviewers' comments:

Reviewer's Responses to Questions

**Comments to the Author**

1. Is the manuscript technically sound, and do the data support the conclusions?

Reviewer #1: Yes

Reviewer #2: Yes

2. Has the statistical analysis been performed appropriately and rigorously?

Reviewer #1: N/A

Reviewer #2: Yes

3. Have the authors made all data underlying the findings in their manuscript fully available?

Reviewer #1: Yes

Reviewer #2: Yes

4. Is the manuscript presented in an intelligible fashion and written in standard English?

Reviewer #1: Yes

Reviewer #2: Yes

Reviewer #1: Dear Editor,

The present manuscript represents a valuable contribution to herpetology, documenting a new species of glassfrog (genus Nymphargus) from Ecuador. The study is well-supported by good data and thorough documentation. However, while the scientific content is commendable, the manuscript suffers from significant issues in presentation. Numerous grammatical and stylistic errors detract from the clarity and professionalism of the work. A comprehensive revision is necessary to improve the formal structure, language, and overall readability of the manuscript before it can be considered for publication.

Also, my main concern is that the authors did not present or discuss any biogeographical evidence to further support the validity of the new species, assuming such information is available. Although the molecular and morphological data are compelling, the new species shows striking similarity to some of its congeners. Therefore, any additional evidence, particularly biogeographical data, would be valuable. I recommend that the authors include biogeographical information, such as geographic isolation, habitat specificity, or distributional gaps, to strengthen the case for recognizing this taxon as a distinct species.

Bellow I present my detailed comments, suggestions and corrections:

32–33: Repetitive. The authors already mention above that N. buenaventura belongs to that clade.

34: I strongly recommend that the authors carefully review the entire manuscript to ensure consistent use of the name for the collection site. The site is referred to variously as “El Quimi,” “El Quimi Biological Reserve,” and other variants. To maintain clarity and avoid confusion, please choose a single standardized name, preferably Reserva Biológica El Quimi, and use it consistently throughout the manuscript

58: This is okay to present the taxonomic authorities for each of the species when first mentioned in the document, however, please ensure consistency by using the same formatting style throughout the entire manuscript.

120: The manuscript refers to “S1 Text,” but this designation is unclear. Based on the current supplementary materials, you have an “S1 Table” and an “S1 Appendix.” If “S1 Text” is meant to refer to the appendix, please revise the terminology throughout the document for consistency. Alternatively, you may consider using “Appendix 1” within the Supplementary Material section. Please ensure that all references to supplementary files are clearly labeled and consistently formatted.

128: Delete “are”.

136: The bioacoustic analysis requires substantial revision. The methodology section is currently incomplete and lacks essential details, such as environmental conditions during recordings and analytical procedures (parameters measured and its definitions, DFT? dB filter bandwidth? etc.). Although the authors cite the work of Köhler et al., they do not follow the fundamental recommendations outlined in that reference regarding the presentation of bioacoustic analyses. To ensure methodological rigor and comparability, I strongly suggest that the authors revisit Köhler et al.'s guidelines and revise their bioacoustic section accordingly. Also, the authors cite the Raven software by referencing its user manual, which is not the appropriate way to cite this tool in a scientific publication.

151: See comments from 58.

165.Table1: add sp. nov. after the name of the new species.

177–181: What is this? Please check the entire manuscript for similar errors.

185–186: Please add a reference for this.

189: See comments from 58.

241–248: This entire paragraph is redundant and unnecessary, as all the relevant information is presented below. I recommend removing it.

258: What does 'SC' refer to? If it is a field or specimen number, please specify this clearly.

258: What is “zona de amortiguamiento”? If it´s buffer zone (as presented below) please correct it.

262: Please specify how many paratypes are.

266: Please include a brief explanation of the grammatical structure of the proposed name, specifying whether it is a noun in apposition, an adjective, or another form, in accordance with the conventions of Latin nomenclature.

289: Please delete the extra space after “half”.

293: What do you mean by 'medium body size'? Is this in reference to glass frogs in general, or specifically to species within the genus Nymphargus? Please clarify.

296: See comments from 58.

299: See comments from 293.

300: Once the new species has been formally defined, please ensure consistency by removing all instances of 'sp. nov.' following the species name throughout the manuscript (except of course in figure captions).

351: The sentence “Dentigerous processes of vomer absent without teeth” is unclear and awkwardly phrased. Please rephrase it.

356–357: You have already specified that Figure 3 shows a preserved specimen, so it is not necessary to repeatedly add “of preserved individual”. Please consider removing these redundant mentions.

365–367: The same as above, but this time … it´s alive. Please rephrase it.

389: Please correct “colouration” to “coloration” and review the entire manuscript to ensure consistency with American English spelling, as required by PLOS ONE.

400–403: You introduced abbreviations for the measurements in the Methods section, but they are not consistently used throughout the manuscript. Please ensure that the abbreviations are applied wherever relevant to maintain consistency and clarity, like in this case.

406–407: Unclear. Please rephrase it.

426: As commented above at 136, the bioacoustic analysis requires substantial revision. Some of the important comments:

- The authors mention in the methods section that the “original recordings are deposited in the audio archive of the QCAZ-A collection”. Does this archive assign identification codes to the recordings? If so, please specify them, as it is essential to link the acoustic analyses to the corresponding archived records for transparency and reproducibility.

- The call consists of “one click”? Could you clarify what type of call this is? It would be helpful to specify whether it is a single-pulsed note, a short tonal element, or another type of vocalization. If “click” refers to the sound quality, please describe it accordingly, for example, that the call resembles a click in its acoustic characteristics.

- Be careful that the call rate should be provided as calls per minute. Please correct it.

- What about the 90% bandwidth or if the Fundamental frequency is visible? Please check carefully Köhler et al.'s guidelines and revise the bioacoustic section accordingly.

Fig. 8: This needs some work because it is too sloppy. Raven allows for the preparation of high-quality sonograms. I recommend removing all unnecessary visual elements such as Position Markers, Selection Control Points, and other interface artifacts before exporting the image. Additionally, it is not necessary to mark the call or note directly on the sonogram to indicate the source of the power spectrum. I strongly encourage the authors to present a clean and professionally edited version of the sonogram. For guidance, please refer to Köhler et al.'s publications, which provide excellent examples of properly formatted sonograms."

It is unclear what panels A, B, C, and D specifically represent. Do A and B show two consecutive calls, while C and D provide a detailed view of a single call? As currently presented, the panels appear to convey overlapping information. Please clarify the purpose of each panel and consider reorganizing or simplifying the figure to enhance its interpretability and relevance.

434–435: It is not necessary to specify that these are oscillograms or spectrograms, as this is visually evident. However, it is important to clearly state what these visualizations represent.

440: See comments from 300 and 34.

444–445: “The type locality is covered by …”?? Please correct the sentence.

468: Discovery or description? As bellow the authors comment that four new species are in the process of description, please clarify the sentence by indicating how many where found and how many were already described.

Reviewer #2: This is a solid description of a new species of Nymphargus from the Cordillera del Condor of Ecuador. Although the authors did not include a comparison of calls with closely related species, the phylogeny strongly supports that this species is new, and thus I don't see this as problematic. Overall, the arguments presented are sufficient to justify the description of this species, and as such I recommend that this paper is published following some minor revisions, detailed below.

Title: from "the" Cordillera del Condor? I saw this in one or two other places in the manuscript (omitting the "the" before Cordillera), so please check throughout.

Line 29: wileyi

Lines 93-101: The notion of a historical fate seems like a strange thing to invoke here, regardless of whether it is part of an existing species concept. For one, "historical fate" is an oxymoron: history is in the past, and fate is in the future, which just doesn't make sense. Second, you cannot know the fate of a lineage, i.e., it is not something that can be determined, only guessed, except maybe in the case of extinct species. Third, and most importantly (in my mind) two fully isolated "good" species could fuse back to a single species in the future. Aside from the impossibility of knowing whether this will happen or not, this means our current interpretation of species limits is to be dictated by some ultimate outcome. Imagine a scenario where two species diverge, we take a "snapshot" at some stage of this divergence, and the two species never fuse. In this case -- with the gift of hindsight-- we can say that we were correct in our inference that they were good species. Now let's "replay life's tape" so to speak, and run this thought experiment a second time. This time, let's say that due to whatever reason, these two populations are brought back into contact and fuse into one. In this case, the inference would be that not only are they ultimately a single species, but also that they should never have been treated as two species in the first place because they did not maintain their "historical fate". Our "snapshot" of the divergence process in both cases is identical, the only thing that varied is the future outcome. This would suggest that species limits in these two cases were in fact unknowable at the time of the snapshot. If I am off-base here due to a misunderstanding of the definition of "historical fate", then this term needs to be clarified here. Also, I realize that this species concept was not written by these authors, so my criticism here is not really targeted at the current study. Still, as the authors endorse and quote this species concept, I think my comments are still relevant. Operationally I doubt it matters as I think most species concepts would recognize this species as new, given the phylogeny.

Lines 177-180: A url seems to have been erroneously pasted here.

Phylogenetics methods: Was an outgroup used? If not, how was the tree rooted? Related to this, none of the splits mentioned in the time-tree section are in Figure 1, so it is not clear how they would be informative here unless there were many more samples that were included in the phylogenetic analysis. Based on what is written and presented, it sounds like the phylogenetic analysis was done exclusively on Nymphargus. Upon downloading the alignment, it is clear than many other non-Nymphargus terminals were included. So this needs to be mentioned in the methods and a Genbank table for all terminals should be included as well.

Line 221: "latter". I also noticed the same error on line 230, so check this throughout.

Figure 2: I would suggest putting the legend on the figure itself, because cross referencing the dots with the figure legend is tedious. Make it easy for the reader. Furthermore, I think a yellow star is the wrong choice for a species that is not the focus of the paper. When I see yellow star on a map, my first thought is type locality of the new species.

Line 292: Include units (presumably Hz)

Lines 295--300: Presumably it is the combination of these characters that distinguish the new species from all but these seven species? If so, clarify that this is "in combination".

Line 300: All Nymphargus with the exception of grandisonae lack humeral spines, right? If so, I do not think the absence of humeral spines should be mentioned here because it would only exclude grandisonae (unless I am mistaken), which is already excluded by the fact that it has dorsal markings. Actually I think this section should be clarified a bit. I think this list of characters should be minimal, that is, if these are the only 7 species that have a uniformly green dorsum without pale or dark marks, then that is the only character that needs to be mentioned. Ok, I think I see the issue now on re-read: this list is for the whole family Centrolenidae. It could simplify things a bit to open this section with a statement regarding the generic placement in Nymphargus, and then write "N. dajomesae is diagnosed from all other species of *Nymphargus* except for XYZ species by a uniformly green dorsum etc..." I think generic placement + diagnosis against Nymphargus is enough here. The generic placement does not have to be purely morphological and can also reference the phylogenetic results, as you included representatives of many other genera in the analysis.

Line 329: Do you mean shagreen without warts? (Instead of just "shagreen"). If so, specify.

Lines 426-427: Is it not possible to provide comparisons to other species? The most logical would be to cariticommatus. My guess is that there is simply not much published regarding the calls of other Nymphargus. If this is the case, it should be mentioned here, because otherwise the reader will be wondering why calls were not compared among species.

Lines 446-449: It would be great to include a figure depicting the habitat and maybe the surrounding landscape.

Line 473: N. lasgralarias and N. sucre should also be mentioned as "additional" species in this clade. In other words this clade is only supported if you also expand it to include additional species (which were not described at the time that this clade was first proposed).

**Do you want your identity to be public for this peer review?** For information about this choice, including consent withdrawal, please see our Privacy Policy

Reviewer #1: No

Reviewer #2: No

---

## [Author Response · Author response to Decision Letter 1]

13 Dec 2025

Response to Reviewers' comments:

Reviewer #1: Dear Editor,

The present manuscript represents a valuable contribution to herpetology, documenting a new species of glassfrog (genus Nymphargus) from Ecuador. The study is well-supported by good data and thorough documentation. However, while the scientific content is commendable, the manuscript suffers from significant issues in presentation. Numerous grammatical and stylistic errors detract from the clarity and professionalism of the work. A comprehensive revision is necessary to improve the formal structure, language, and overall readability of the manuscript before it can be considered for publication.

Also, my main concern is that the authors did not present or discuss any biogeographical evidence to further support the validity of the new species, assuming such information is available. Although the molecular and morphological data are compelling, the new species shows striking similarity to some of its congeners. Therefore, any additional evidence, particularly biogeographical data, would be valuable. I recommend that the authors include biogeographical information, such as geographic isolation, habitat specificity, or distributional gaps, to strengthen the case for recognizing this taxon as a distinct species.

We thank the reviewer for this comment. We have implemented this recommendation by adding a paragraph in the Discussion section where we hypothesize a speciation scenario based on the known geographic distribution of the new species and its sister.

Bellow I present my detailed comments, suggestions and corrections:

32–33: Repetitive. The authors already mention above that N. buenaventura belongs to that clade.

We agree with the reviewer and followed his/her recommendation by deleting the redundant section.

34: I strongly recommend that the authors carefully review the entire manuscript to ensure consistent use of the name for the collection site. The site is referred to variously as “El Quimi,” “El Quimi Biological Reserve,” and other variants. To maintain clarity and avoid confusion, please choose a single standardized name, preferably Reserva Biológica El Quimi, and use it consistently throughout the manuscript

Thank you for this comment, we changed the site to “Reserva Biológica El Quimi” throughout the manuscript.

58: This is okay to present the taxonomic authorities for each of the species when first mentioned in the document, however, please ensure consistency by using the same formatting style throughout the entire manuscript.

We corrected the citations following the References guidelines throughout the manuscript.

120: The manuscript refers to “S1 Text,” but this designation is unclear. Based on the current supplementary materials, you have an “S1 Table” and an “S1 Appendix.” If “S1 Text” is meant to refer to the appendix, please revise the terminology throughout the document for consistency. Alternatively, you may consider using “Appendix 1” within the Supplementary Material section. Please ensure that all references to supplementary files are clearly labeled and consistently formatted.

We checked and corrected the labels following the respective guidelines.

128: Delete “are”.

The word “are” has been deleted as suggested.

136: The bioacoustic analysis requires substantial revision. The methodology section is currently incomplete and lacks essential details, such as environmental conditions during recordings and analytical procedures (parameters measured and its definitions, DFT? dB filter bandwidth? etc.). Although the authors cite the work of Köhler et al., they do not follow the fundamental recommendations outlined in that reference regarding the presentation of bioacoustic analyses. To ensure methodological rigor and comparability, I strongly suggest that the authors revisit Köhler et al.'s guidelines and revise their bioacoustic section accordingly. Also, the authors cite the Raven software by referencing its user manual, which is not the appropriate way to cite this tool in a scientific publication.

We thank the reviewer for this useful comment. We added the missing parameters in the Methodology section. We also corrected the reference for the software.

151: See comments from 58.

These are not citations; these are specimen numbers in parenthesis

165.Table1: add sp. nov. after the name of the new species.

The term “sp. nov” has been added.

177–181: What is this? Please check the entire manuscript for similar errors.

The erroneous URL was removed.

185–186: Please add a reference for this.

We thank the reviewer for this helpful comment. The reference has been added.

189: See comments from 58.

These are not citations, it is additional information in parenthesis

241–248: This entire paragraph is redundant and unnecessary, as all the relevant information is presented below. I recommend removing it.

We thank the reviewer for this valuable observation. The paragraph has been removed as suggested.

258: What does 'SC' refer to? If it is a field or specimen number, please specify this clearly.

It is a field number, but we deleted it to avoid confusions

258: What is “zona de amortiguamiento”? If it´s buffer zone (as presented below) please correct it.

The term “Zona de amortiguamiento” has been replaced with “buffer zone” as suggested.

262: Please specify how many paratypes are.

We appreciate the reviewer’s comment. The number of paratypes has been specified in the text as follows: “Paratypes: 7 adult males…”

266: Please include a brief explanation of the grammatical structure of the proposed name, specifying whether it is a noun in apposition, an adjective, or another form, in accordance with the conventions of Latin nomenclature.

We thank the reviewer for this useful comment. We have clarified the grammatical nature of the species name in the Etymology section.

289: Please delete the extra space after “half”.

The extra space after “half” has been removed

293: What do you mean by 'medium body size'? Is this in reference to glass frogs in general, or specifically to species within the genus Nymphargus? Please clarify.

As indicated in the section “Morphology and coloration” in Materials and Methods, all aspects related to diagnosis, terminology, and measurements follow the format and definitions proposed by Cisneros-Heredia & McDiarmid [5]. The reference to “medium body size” corresponds to the size categories specified in that work.

296: See comments from 58.

We corrected as suggested.

299: See comments from 293.

As indicated in the section “Morphology and coloration” in Materials and Methods, all aspects related to diagnosis, terminology, and measurements follow the format and definitions proposed by Cisneros-Heredia & McDiarmid [5]. The reference to “medium body size” corresponds to the size categories specified in that work.

300: Once the new species has been formally defined, please ensure consistency by removing all instances of 'sp. nov.' following the species name throughout the manuscript (except of course in figure captions).

All instances of “sp. nov” have been removed throughout the manuscript as suggested.

351: The sentence “Dentigerous processes of vomer absent without teeth” is unclear and awkwardly phrased. Please rephrase it.

Thanks, it should be only “Dentigerous processes of vomer absent”.

356–357: You have already specified that Figure 3 shows a preserved specimen, so it is not necessary to repeatedly add “of preserved individual”. Please consider removing these redundant mentions.

Redundant mentions have been removed.

365–367: The same as above, but this time … it´s alive. Please rephrase it.

Redundant mentions have been removed.

389: Please correct “colouration” to “coloration” and review the entire manuscript to ensure consistency with American English spelling, as required by PLOS ONE.

We corrected from “colouration” to “coloration” in the entire manuscript.

400–403: You introduced abbreviations for the measurements in the Methods section, but they are not consistently used throughout the manuscript. Please ensure that the abbreviations are applied wherever relevant to maintain consistency and clarity, like in this case.

We thank the reviewer for the observation. The use of measurement abbreviations has been revised throughout the manuscript.

406–407: Unclear. Please rephrase it.

The sentence has been rephrased for clarity

426: As commented above at 136, the bioacoustic analysis requires substantial revision. Some of the important comments:

- The authors mention in the methods section that the “original recordings are deposited in the audio archive of the QCAZ-A collection”. Does this archive assign identification codes to the recordings? If so, please specify them, as it is essential to link the acoustic analyses to the corresponding archived records for transparency and reproducibility.

We thank the reviewer for this comment. We added the required clarification in the Methodology section.

- The call consists of “one click”? Could you clarify what type of call this is? It would be helpful to specify whether it is a single-pulsed note, a short tonal element, or another type of vocalization. If “click” refers to the sound quality, please describe it accordingly, for example, that the call resembles a click in its acoustic characteristics.

We have added the type of call following the reviewer recommendation.

- Be careful that the call rate should be provided as calls per minute. Please correct it.

We agree with the reviewer and followed his/her recommendation.

- What about the 90% bandwidth or if the Fundamental frequency is visible? Please check carefully Köhler et al.'s guidelines and revise the bioacoustic section accordingly.

We thank the reviewer for this helpful comment. We have made the requested changes.

Fig. 8: This needs some work because it is too sloppy. Raven allows for the preparation of high-quality sonograms. I recommend removing all unnecessary visual elements such as Position Markers, Selection Control Points, and other interface artifacts before exporting the image. Additionally, it is not necessary to mark the call or note directly on the sonogram to indicate the source of the power spectrum. I strongly encourage the authors to present a clean and professionally edited version of the sonogram. For guidance, please refer to Köhler et al.'s publications, which provide excellent examples of properly formatted sonograms."

We thank the reviewer for this comment. We cleaned up the Figure from unnecessary visual elements. We also took to opportunity to improve the power spectrum, oscillogram and spectrogram of the single call. We also added labels for the harmonics.

It is unclear what panels A, B, C, and D specifically represent. Do A and B show two consecutive calls, while C and D provide a detailed view of a single call? As currently presented, the panels appear to convey overlapping information. Please clarify the purpose of each panel and consider reorganizing or simplifying the figure to enhance its interpretability and relevance.

We thank the reviewer for this comment. We added the required clarification.

434–435: It is not necessary to specify that these are oscillograms or spectrograms, as this is visually evident. However, it is important to clearly state what these visualizations represent.

We thank the reviewer for this comment. However, we would prefer to leave the specification of the type of graph. Clearly, the reviewer has a deep knowledge of bioacoustic analyses. Unfortunately, many potential readers of this article may not be as knowledgeable. We will prefer to leave the types of sound representations for them.

440: See comments from 300 and 34.

The changes were made as suggested.

444–445: “The type locality is covered by …”?? Please correct the sentence.

The sentence has been revised and corrected. The sentence now reads as: “The type locality is within Eastern Montane Forest (sensu [25]) or Evergreen Montane Forest on sandstone plateaus of Cordillera del Cóndor (sensu [68]).”

468: Discovery or description? As bellow the authors comment that four new species are in the process of description, please clarify the sentence by indicating how many where found and how many were already described.

We thank the reviewer for this helpful comment. We have expanded this section to state more precisely how many species were described and how many species were new. By digging in our databases we realized that the pattern was quite different between the high plateau and the adjacent lowlands. We added a new paragraph reporting the notable difference.

Reviewer #2: This is a solid description of a new species of Nymphargus from the Cordillera del Condor of Ecuador. Although the authors did not include a comparison of calls with closely related species, the phylogeny strongly supports that this species is new, and thus I don't see this as problematic. Overall, the arguments presented are sufficient to justify the description of this species, and as such I recommend that this paper is published following some minor revisions, detailed below.

Title: from "the" Cordillera del Condor? I saw this in one or two other places in the manuscript (omitting the "the" before Cordillera), so please check throughout.

We thank the reviewer for this helpful comment. We have checked the manuscript to consistently use “Cordillera del Cóndor” without “the”.

Line 29: wileyi

We corrected from “wiley” to “wileyi”.

Lines 93-101: The notion of a historical fate seems like a strange thing to invoke here, regardless of whether it is part of an existing species concept. For one, "historical fate" is an oxymoron: history is in the past, and fate is in the future, which just doesn't make sense. Second, you cannot know the fate of a lineage, i.e., it is not something that can be determined, only guessed, except maybe in the case of extinct species. Third, and most importantly (in my mind) two fully isolated "good" species could fuse back to a single species in the future. Aside from the impossibility of knowing whether this will happen or not, this means our current interpretation of species limits is to be dictated by some ultimate outcome. Imagine a scenario where two species diverge, we take a "snapshot" at some stage of this divergence, and the two species never fuse. In this case -- with the gift of hindsight-- we can say that we were correct in our inference that they were good species. Now let's "replay life's tape" so to speak, and run this thought experiment a second time. This time, let's say that due to whatever reason, these two populations are brought back into contact and fuse into one. In this case, the inference would be that not only are they ultimately a single species, but also that they should never have been treated as two species in the first place because they did not maintain their "historical fate". Our "snapshot" of the divergence process in both cases is identical, the only thing that varied is the future outcome. This would suggest that species limits in these two cases were in fact unknowable at the time of the snapshot. If I am off-base here due to a misunderstanding of the definition of "historical fate", then this term needs to be clarified here. Also, I realize that this species concept was not written by these authors, so my criticism here is not really targeted at the current study. Still, as the authors endorse and quote this species concept, I think my comments are still relevant. Operationally I doubt it matters as I think most species concepts would recognize this species as new, given the phylogeny.

We thank the reviewer for this comment. The discussion on the merits of different species concepts is fascinating but goes beyond the scope of our manuscript. We agree with the reviewer in that most species concepts would recognize N. dajomesae as new. We followed her/his recommendation by clarifying the meaning of “historical fate” in the methodology section.

Lines 177-180: A url seems to have been erroneously pasted here.

The erroneous URL was removed.

Phylogenetics methods: Was an outgroup use

---

## [Editor Report · Decision Letter 1]

14 Jan 2026

I invite you to submit a revised version of the manuscript that addresses the few (very minor) points raised in the attached document. Please make sure to re-upload a corrected Figure 2 (see specific comment in the attachment).

We look forward to receiving your revised manuscript.

Kind regards,

Philippe J. R. Kok, Ph.D.

Academic Editor

PLOS One
---

## [Author Response · Author response to Decision Letter 2]

26 Feb 2026

From the Decision letter: "I invite you to submit a revised version of the manuscript that addresses the few (very minor) points raised in the attached document. Please make sure to re-upload a corrected Figure 2 (see specific comment in the attachment)."

We have made all the few minor points raised to the previous version. A point by point response is shown in the manuscript with tracked changes (as responses to comments).

Dear Editor, we implemented all the required changes. Details are listed below:

RESPONSE:

• “This sounds odd. Wouldn't it make more sense to just mention that all authors contributed equally?”

We thank the reviewer for this comment. We have followed his/her recommendation.

• “Please replace the current Spanish text within the figure with "Type locality"”

The requested change was made. The new version of the Figure is being uploaded.

• “I count six in the list you provide, please correct where necessary”

Thanks for noticing this error. It has been fixed.

• “Any temperature data?”

Unfortunately, temperature data is unavailable. We added a statement with that clarification in the methodology section.

---

## [Editor Report · Decision Letter 2]

2 Mar 2026

A secret from a hidden world: A new glassfrog of the genus Nymphargus (Anura: Centrolenidae) from Cordillera del Cóndor, Ecuador

PONE-D-25-51397R2

Dear Dr. Ron,

We’re pleased to inform you that your manuscript has been judged scientifically suitable for publication and will be formally accepted for publication once it meets all outstanding technical requirements.

Kind regards,

Philippe J. R. Kok, Ph.D.

Academic Editor

PLOS One
---

## [Editor Report · Acceptance letter]

PONE-D-25-51397R2

PLOS One

Dear Dr. Ron,

I'm pleased to inform you that your manuscript has been deemed suitable for publication in PLOS One. Congratulations! Your manuscript is now being handed over to our production team.

Kind regards,

on behalf of

Prof. Philippe J. R. Kok

Academic Editor

PLOS One